# Cilioretinal Artery Occlusion after Endovascular Coil Embolization for Anterior Communicating Artery

**DOI:** 10.3390/brainsci11050542

**Published:** 2021-04-25

**Authors:** Heewon Bae, TaeGu Kang, Da-Eun Jeong, KyuHwan Shim, MinJu Kang

**Affiliations:** 1Department of Neurology, Veterans Health Service Medical Center, 53 Jinhwangdo-ro 61-gil, Gangdong-gu, Seoul 05368, Korea; communis@nate.com (H.B.); doctorjung86@gamil.com (D.-E.J.); smuller0305@gamil.com (K.S.); 2Department of Ophthalmology, Veterans Health Service Medical Center, 53 Jinhwangdo-ro 61-gil, Gangdong-gu, Seoul 05368, Korea; raingrapher@hanmail.net

**Keywords:** coil embolization, cilioretinal artery occlusion, branch retinal artery infarction

## Abstract

Unruptured intracranial aneurysms have a risk of rupture, so coil embolization is widely practiced as it preserves a patent artery. There are complications of coil procedures, such as patent artery occlusion and thromboembolism, which can result in retinal artery occlusion. We report onretinal artery occlusion following coil embolization of anterior communicating artery aneurysm. This is a rare case of a combination of cilioretinal and branch retinal artery occlusion, and is unusual in showing a functional recovery.

## 1. Introduction

Unruptured intracranial aneurysms occur in about 1–5% of the adult population, with a relatively high probability of anterior communicating artery aneurysms (ACoA) (30%) [1,2]. An intracranial aneurysm 7 mm or larger has a high risk of rupture that can be prevented by surgical clipping or endovascular coiling [3]. Coil embolization is widely practiced due to number of advantages, such as minimal invasiveness and high success rate, compared to other procedures [3]. However, there are several complications after the procedure related to ischemic events. Cerebrovascular complications, including patent artery narrowing, granuloma formation, and bleeding from unruptured aneurysms, can occur [4]. Thromboembolism is a major procedural complication and may result in retinal infarction in a small number of cases [5].

However, there are few reports of retinal artery occlusion caused by coil embolization of ACoA aneurysms located in distal to the ophthalmic artery.We report a case in which a patient with cilioretinal artery and branch retinal artery occlusion after ACoA aneurysm procedure showed functional recovery.

## 2. Case

A 72-year-old man presented one month ago with a sudden pulsating continuous headache with anumeric rating scalescore of 8. Brain magnetic resonance imaging and brain magnetic resonance angiography were performed to identify structural lesions. A cerebral aneurysm of 6.2 mm in maximal diameter was detected in the anterior communicating artery (ACoA), and the patient was transferred to the neurosurgery for coil embolization. Cerebral angiography performed before the procedure found no stenosis or sign of vasospasmof the carotid and intracranial arteries. Coil embolization was successfully completed (Figure 1). However, the day after the procedure, the patient complained that a small fixed dark gray square was visible on the inferior central area and felt that hisvision was out of focus. The confrontation test did not show prominent visual field defects. On anterior segment examination, no abnormality was observed in both eyes. Intraocular pressure (IOP) measured using non-contact tonometry was 15/16, the corrected visual acuity was 20/20 in the right eye and 20/63 in the left eye, and pupillary light reflexes were intact. The retina showed partial whitening (0.5 DD) in the perimacular area at the 12 o’clock position, indicating the possibility of cilioretinal artery occlusion (CLRAO). Moreover, there was a branch retinal artery occlusion 1 DD (disc diameter) away from the optic disc to the inferior temporal area, accompanied by inner retinal edema, and macular pigment epithelial detachment (PED) was detected.

The diagnosis was cilioretinal artery occlusion with branch retinal artery occlusion (BRAO). Two weeks later, the IOP measured in both eyes was 14/13, visual acuity was 20/20 in the right eye, 20/63 in the left eye, with slight difference tobefore. Optical coherence tomography (OCT) showed that the edema remained, but the size of PED slightly decreased.

At a one month follow-up, there was no change in the IOP, but the patient’s vision improved to 20/32 in the left eye, compared with the previous examinations. Fundus examination showed reduced ischemic lesions in the perimacular area, and the overall artery flexus defect showed a slight improvement in OCT angiography, but some ischemic lesions remained. Tests performed using the Humphrey visual field analyzer (30-2 program) showed preserved function in the perimacular area. Two months later, visual acuity was still 20/32 in the left eye, but ischemic lesions in the retina recovered almost completely. The capillary perfusion also improved in OCT angiography, and the patient is currently under outpatient follow-up care (Figure 2 and Figure 3).

## 3. Discussion

Coil embolization of cerebral aneurysms is widely practiced as it preserves a patent artery and can be safely performed in a short time [6]. The most common complications of the procedure are patent artery occlusion (2%) and thromboembolism (2.4%) [3]. Thromboembolism mainly causes cerebral infarction but can cause retinal infarction when the thrombus develops in the ophthalmic artery. The main symptoms of retinal artery occlusion are blurred vision or visual field defects. However, our patient presented only a non-specific symptom of localized blurred vision near the macula, making it difficult to consider the possibility of retinal infarction.

An ophthalmic examination revealed the presence of combined CLRAO and BRAO. The cilioretinal artery occurs as an anatomic variant in 18.5% of the population and supplies oxygen to the macula. It is a branch of the short posterior ciliary artery and rarely causes ischemic lesions. A study by Heller et al. found that 3 of 104 patients treated with coil embolization had CRAO, and BRAO alone was reported in several cases [7,8,9]. Notably, this is the first case in which CLRAO and BRAO appeared together after coil embolization.

There are several reasons for the occurrence of retinal infarction. Retinal artery occlusion, primarily caused by procedure itself, is a possibility. Use of catheters, direct contact with the wall of the intracranial artery, or carotid plaque break off while passing the catheter may develop thrombus caused ophthalmic artery occlusion during surgery [10]. However, cerebral angiographyconfirmed that the carotid artery was intact, and hence, the risk was not high. The event occurred in the anterior communicating artery aneurysm procedure distal to the ophthalmic artery, but considering the direction of the blood flow, it was unlikely that the thrombus caused RAO.

The microembolism that caused retinal infarction may be related to air embolism during the injection of heparin into the catheter [1]. It is worth noting that CLRAO and BRAO are multifocal lesions, which may have been caused by decreased blood flow due to vasospasms. Vasospasms may result from mechanisms such as structural changes in the arterial wall, breakdown of blood products, or inflammation due to catheter-induced stimulation of the endothelium [11,12]. Moreover, vasospasm may have caused ischemic lesions in the previous retinal artery stenosis.

In our case, retinal infarction functionally recovered after a month, with improvements in central vision and Humphrey visual field test results. We also confirmed a reduction in ischemic lesions. These findings are not usually observed in patients with retinal artery occlusion, suggesting that they may have been caused by reperfusion of an embolism or recovery fromvasospasm.

## 4. Conclusions

This case is significant for several reasons. First, as the symptoms are very non-specific, retinal infarction should not be ruled out, even in the absence of blurred vision or visual field defects. Second, previous case reports have described coil embolization near the ophthalmic artery, but this is the first report of RAO following coil embolization of ACoA aneurysm. It is also noteworthy that combined CLRAO and BRAO, which is rare, showed functional recovery. As there is a possibility of ophthalmic complications after the coil procedure, an ophthalmological examination is necessary, even if there are unusual symptoms. Furthermore, this case shows that the likelihood of functional recovery should be considered in retinal artery occlusion following coil embolization.

## Figures and Tables

**Figure 1 brainsci-11-00542-f001:**
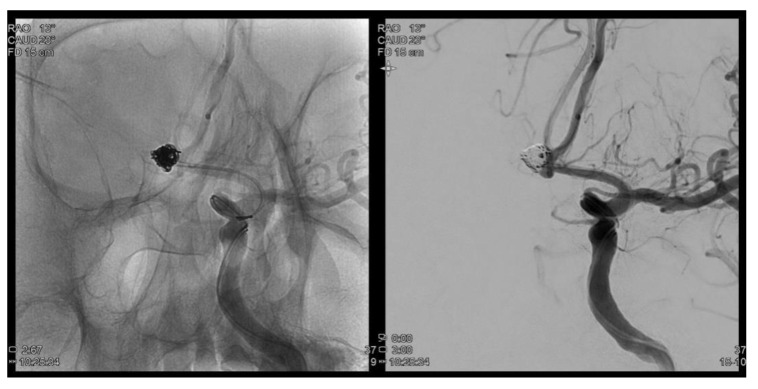
Trans femoral cerebral angiography images show a 6.2 mm sized AcomA (anterior communiating artery) aneurysm and AcomA aneurysm was embolized following coil procedure.

**Figure 2 brainsci-11-00542-f002:**
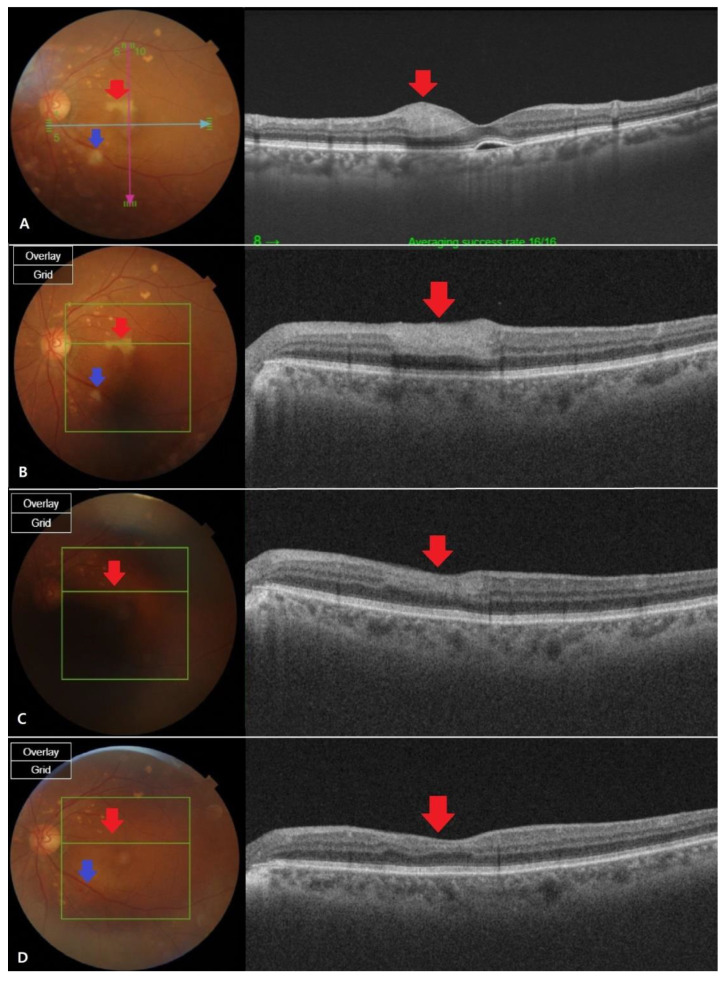
Fundus photography and optical coherence tomography (OCT) images at different stages of retinal artery occlusion (RAO). (**A**) The fundus findings were confirmed to be cilioretinal artery occlusion (CLRAO) by observing whitening and intraretinal thickness in the 12 o’clock direction near the macular on the day the symptoms occurred.There is a cotton wool patchfinding in the disc’s inferotemporal area, which is one disc dimeter away, showing branch retinal artery occlusion (BRAO). (**B**) After two weeks, the apparent finding of the whitening was not recovered (red arrow), but the intraretinal thickness of the OCT decreased.It is thought that the area of BRAO was reduced due to reperfusion (blue arrow). (**C**) After a month, the fundus found that the whitening area recovered and the intraretinal thickness was completely reduced in OCT, which was in normal range (red arrow). BRAO area was almost recovered, but still a little whitening was left (blue Arrow). (**D**) Two months later, full recovery was shown from the fundus findings (red arrow). Mild intraretinal thinning was shown in OCT (blue arrow), but the BRAO recovered so that it was almost indistinguishable by visual observation.

**Figure 3 brainsci-11-00542-f003:**
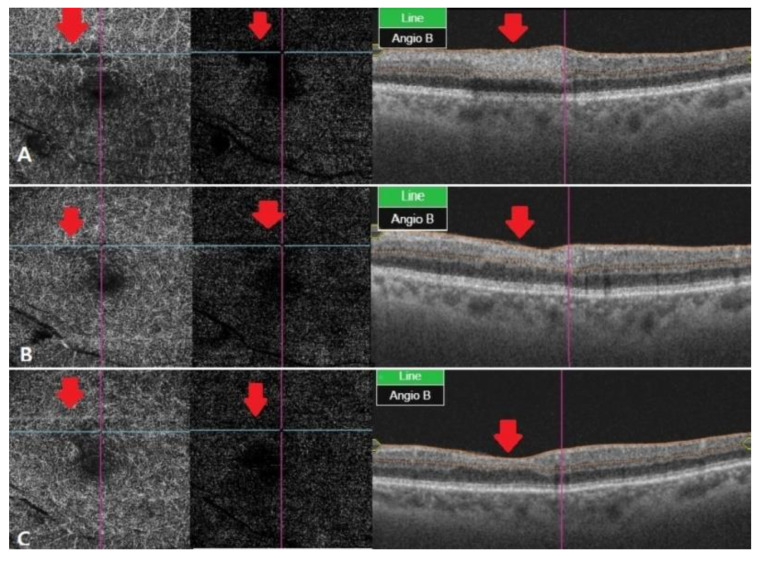
OCT angiography at different stages of RAO. (**A**) After two weeks, OCT angiography was observed with vessel damage and window defect in deep and outer retina. (**B**) After a month, OCT angiography showed recovered vascular damage of deep and outer retina. This seems to be due to reperfusion and reduced window defect. (**C**) Two months later, deep and outer retinafully recovered in vascular findings. The macular window defect in the choriocapillaris area seems to be due to underlying drusen and PED.

## Data Availability

Not applicable.

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
