# Peer review of "Cilioretinal Artery Occlusion after Endovascular Coil Embolization for Anterior Communicating Artery"

_brainsci, 2021, doi:10.3390/brainsci11050542_

Round 1

Reviewer 1 Report

Authors reported a rare case of a combination of cilioretinal and branch retinal artery occlusion and functional recovery. Please look a these points:

  1. "The complication that occurs after procedure can result in retinal infarction caused by thromboembolism" This sentece is uncorrect, what does it mean? It seems that the only complication of endovascular treatment is retinal infarction. Please revise.
  2. Introduction is very short. Although the paper is only a case report, it is necessary to better introduce the issue of endovascular and surgical complications of aneurysms,  including parent artery narrowing, granuloma formation, fatal bleeding and thromboembolism, Please look at these important references:  Wrapping of intracranial aneurysms: Single-center series and systematic review of the literature. Br J Neurosurg. 2015;29(6):785-91. doi: 10.3109/02688697.2015.1071320.           Differences in thromboembolism after stent-assisted coiling for unruptured aneurysms between aspirin plus clopidogrel and ticagrelor. J Clin Neurosci. 2020 Dec;82(Pt A):128-133. doi: 10.1016/j.jocn.2020.10.042.           Differences in thromboembolism after stent-assisted coiling for unruptured aneurysms between aspirin plus clopidogrel and ticagrelor. J Clin Neurosci. 2020 Dec;82(Pt A):128-133. doi: 10.1016/j.jocn.2020.10.042. 
  3. "In our case, retinal infarction functionally recovered after a month, with improvements in central vision and Humphrey visual field test results". How the authors can explain that?

Overall an interesting case report.

Author Response

Thank you for your valuable opinions. Please see the attachment.

Reviewer 2 Report

- NRS is not defined in extent
- 6.2 mm size - please specify is the maximal diameter
- small fixed dark gray square - in what field of vision?
Transfemoral cerebral angiography (TFCA) should not be abbreviated since it is used only once again; should be only "cerebral angiography"

The authors point several reasons but the first one is related primarily to the procedure itself and the use of catheters and lesion of the intracranial carotid artery wall. I would say that this is a direct complication of the procedure itself.

Please review the angiography and look for signs of vasospasm? if not please state it. 

How does a thrombus formation in the aneurysm goes back into the ophthalmic artery and cause BRAO? I do think this is a major mechanism

I suggest changing the title to "Cilioretinal Artery Occlusion Occlusion after Endovascular Coil Embolization for Anterior Communicating Artery" because this is the innovative aspect of this case.

Author Response

(The authors gave the same response as above.)

Round 2

Reviewer 1 Report

Authors solved all my criticisms.